# GM1 Oligosaccharide Crosses the Human Blood–Brain Barrier In Vitro by a Paracellular Route

**DOI:** 10.3390/ijms21082858

**Published:** 2020-04-19

**Authors:** Erika Di Biase, Giulia Lunghi, Margherita Maggioni, Maria Fazzari, Diego Yuri Pomè, Nicoletta Loberto, Maria Grazia Ciampa, Pamela Fato, Laura Mauri, Emmanuel Sevin, Fabien Gosselet, Sandro Sonnino, Elena Chiricozzi

**Affiliations:** 1Department of Medical Biotechnology and Translational Medicine, University of Milano, 20122 Milano, Italy; erika.dibiase@unimi.it (E.D.B.); giulia.lunghi@unimi.it (G.L.); maria.fazzari@unimi.it (M.F.); yuri.pome@gmail.com (D.Y.P.); nicoletta.loberto@unimi.it (N.L.); maria.ciampa@unimi.it (M.G.C.); pamela.fato@unimi.it (P.F.); laura.mauri@unimi.it (L.M.); sandro.sonnino@unimi.it (S.S.); 2Blood-Brain Barrier Laboratory, UR2465, Artois University, F-62300 Lens, France; emmanuel.sevin@univ-artois.fr

**Keywords:** ganglioside GM1, GM1-oligosaccharide, blood–brain barrier, brain-like endothelial cells, drug discovery, neurodegeneration, Parkinson’s disease

## Abstract

Ganglioside GM1 (GM1) has been reported to functionally recover degenerated nervous system in vitro and in vivo, but the possibility to translate GM1′s potential in clinical settings is counteracted by its low ability to overcome the blood–brain barrier (BBB) due to its amphiphilic nature. Interestingly, the soluble and hydrophilic GM1-oligosaccharide (OligoGM1) is able to punctually replace GM1 neurotrophic functions alone, both in vitro and in vivo. In order to take advantage of OligoGM1 properties, which overcome GM1′s pharmacological limitations, here we characterize the OligoGM1 brain transport by using a human in vitro BBB model. OligoGM1 showed a 20-fold higher crossing rate than GM1 and time–concentration-dependent transport. Additionally, OligoGM1 crossed the barrier at 4 °C and in inverse transport experiments, allowing consideration of the passive paracellular route. This was confirmed by the exclusion of a direct interaction with the active ATP-binding cassette (ABC) transporters using the “pump out” system. Finally, after barrier crossing, OligoGM1 remained intact and able to induce Neuro2a cell neuritogenesis by activating the TrkA pathway. Importantly, these in vitro data demonstrated that OligoGM1, lacking the hydrophobic ceramide, can advantageously cross the BBB in comparison with GM1, while maintaining its neuroproperties. This study has improved the knowledge about OligoGM1′s pharmacological potential, offering a tangible therapeutic strategy.

## 1. Introduction

Ganglioside GM1 (GM1) has been known for its neurotrophic properties for at least half a century [1], as outlined by several authors in recent years [2,3,4]. GM1 is an integral and fundamental lipid component located on the outer leaflets of membranes of all mammalian cells, and is particularly enriched on the neuronal plasma membrane, where it interacts with the neighboring proteins to modulate intracellular signaling, mainly of biochemical pathways of neuronal differentiation, homeostasis, protection, and restoration [5,6]. GM1′s physiological capacity to recover the functions of damaged central nervous system (CNS) has been widely reported, both in vitro and in vivo [2,3,4]. However, transferring the GM1 neuronal potential from the preclinical models to the clinic depends on its ability to reach the brain structures when administered peripherally. This last aspect is actually severely limited by GM1′s amphiphilicity, which hampers the passage across the blood–brain barrier (BBB) [2,7,8]. This barrier is located at the brain microvessel level and strongly restricts the entry of cells and xenobiotics into the CNS. The BBB anatomical support is the specialized brain endothelial cells of these vessels. These cells do not show any fenestrations and are specifically characterized by the presence of tight junctions (TJ), composed of junctional proteins and TJ-associated proteins. Additionally, these cells express several proteins belonging to the ATP-binding cassette (ABC) transporters family, which act as efflux pumps and play an important role by limiting BBB crossing of several drugs and biological molecules [9,10]. The major efflux pumps expressed at the BBB are P-glycoprotein (P-gp, aka ABCB1), breast cancer resistance protein (BCRP, aka ABCG2), and the multidrug-resistance-associated proteins (MRPs, aka ABCCs).

Importantly, the fate of peripherally administered GM1 observed in the rodent models has not been determined in humans, as often the two species display different BBB permeability rates for the same molecule [7,8]. One of the possible explanations for these discrepancies is the species difference recently reported between rodents and human for the expression of several receptors and transporters expressed in the BBB, including the ABC transporters [11].

In fact, to obtain positively relevant results in parkinsonian patients, a high dose of GM1 has been used in clinical trials in which GM1 was peripherally administered [12], and even for the treatment of patients with Alzheimer’s, GM1 was administered via intracranial injection to obtain a beneficial effect [13], an administration route that is not compatible with an adequate quality of human life.

Many unsuccessful efforts have been made by researchers to find a solution allowing GM1 to reach the brain’s central areas. For example, the soluble alternative of GM1, known as LIGA 20, has been proposed. This molecule is modified in the ceramide structure with a dichloroacetyl group instead of the acyl chain, which although rendering the molecule more permeable, has made it toxic in the long term [14,15,16,17,18,19,20].

Another strategy attempted in rodents employed the intraventricular injection of the *Vibrio cholerae* sialidase, which removes residues of sialic acid from the polysialylated gangliosides (i.e., GD1a), thus increasing the endogenous GM1 levels in the membrane [21]. Unfortunately, even this strategy cannot be performed in the clinic, and peripheral injection of the enzyme would result in it being blocked by the BBB if not properly conveyed to the brain. As reported in recent reviews, many efforts are still ongoing and seem to be concentrated on the possibility of using vehicles (i.e., liposomes) to drive the GM1 into the brain [2,7,8], however without any reported success to date at the best of our knowledge.

In this context, which seems to be disadvantaging the therapeutic use of GM1, our studies have investigated the molecular mechanism underlying the neurotrophic properties of GM1. Indeed, we recently recognized that GM1 exerts its neurotrophic functions by interacting with and activating plasma membrane receptors throughout its oligosaccharide portion—GM1-oligosaccharide (OligoGM1) [22,23,24,25]. We demonstrated that OligoGM1, alone and without entering into the cells, perfectly replicates the neuronal differentiative properties of GM1 in neuroblastoma Neuro2a (N2a) cells and in primary neurons [22,26]. Additionally, OligoGM1 confers protection from 1-Methyl-4-phenyl-1,2,3,6-tetrahydropyridine (MPTP) neurotoxicity and oxidative stress by increasing mitochondrial biogenesis and activity [24,27]. Most importantly, peripherally injected OligoGM1 is able to reach the CNS and to revert the behavioral symptoms and biochemical lesions of a Parkinson’s disease mouse model [25]. OligoGM1, without the hydrophobic ceramide, is chemo-physically different from the entire GM1 (Figure 1)—it is a small hydrophilic molecule, is soluble in aqueous solution, and thus may gain the ability to efficiently access the CNS.

Therefore, we designed this study to improve the knowledge on the OligoGM1 transport across the BBB by employing a recent established human model of brain-like endothelial cells (hBLECs) [28] together with the “pump out” Caco-2 cells model [29]. These experiments show a passive and paracellular passage route through the BBB for the OligoGM1, which importantly results in transport efficiency 20 times greater than that of GM1. Importantly, we establish a reasonable basis for considering the OligoGM1 as an agent that overcomes the GM1 pharmacological limits and could show significant therapeutic benefits for neurodegenerative diseases of the CNS.

## 2. Results

### 2.1. Permeability and Viability of hBLECs

In order to verify the tightness and integrity of the hBLECs, we measured the Lucifer yellow (LY, 50 μM) apparent permeability (Papp) and endothelial permeability (Pe) in relation to its direct passage. Papp and Pe were calculated following Equations (1) and (7)–(10), respectively, as reported in the Appendix A. The obtained permeability values for LY (Pe = 0.65 ± 0.041; Papp = 1.25 ± 0.123) are comparable to the values found previously [28,31,32], demonstrating the integrity of hBLECs and the presence of well-established TJs (Figure 2).

Furthermore, to evaluate the possible toxicity of GM1 and OligoGM1 to the viability of hBLECs, the LY Pe and Papp were assessed in combination with GM1 or OligoGM1 at different concentrations (10, 50, 100, and 300 μM) for 60 min and over time (20, 40, and 60 min). As shown in Figure 2 and in Appendix A, no variations of LY Pe and Papp were observed, even in the presence of OligoGM1 and GM1, suggesting that neither GM1 nor OligoGM1 induce cytotoxic effects on hBLECs. In parallel, as positive control for hBLECs leakage, LY permeability was measured in combination with d-mannitol (1.4 M), which at that concentration induces an osmotic shock, increasing the paracellular permeability [33]. As shown in Figure 2, the co-incubation with d-mannitol is enough to induce a 20-fold and a 4-fold rise of Pe and Papp LY, respectively, as previously demonstrated [33].

Thus, these experiments highlighted the hBLECs’ physiological formation and maintenance in the present study, even in the presence of OligoGM1 or GM1, which do not induce any increase in permeability values.

### 2.2. GM1 and OligoGM1 Transport Study across the hBLECs

The influx of GM1 or OligoGM1 (50 μM plus 1 × 10^6^ dpm) from apical to basolateral sides was evaluated to study their permeability across the hBLECs. As shown in Figure 3, the Pe coefficient for OligoGM1 (0.58 ± 0.051) is comparable to LY at one- (0.65 ± 0.041) and 20-fold higher than GM1 (0.028 ± 0.007). This result suggests that OligoGM1 crosses the hBLEC barrier more efficiently than GM1. Additionally, since LY is a marker molecule of paracellular passive passage across endothelial TJs [28,34] and the Pe values of LY and OligoGM1 are similar, it is plausible to think there may also be a possible paracellular route for OligoGM1.

### 2.3. OligoGM1 Fate in hBLECs

In order to evaluate the possibility for hBLECs to internalize the isotopic tritium-labeled OligoGM1, 50 μM [^3^H]OligoGM1 was administered to the apical side. After 20, 40, 60, and 240 min, the solutions contained in the apical and basolateral sides were collected, the hBLECs-filters were washed with Hepes buffer Ringer’s solution (RH buffer), and the following cells were lysed in order to evaluate the quantity of internalized [^3^H]OligoGM1. At each analyzed time point, about 99% of [^3^H]OligoGM1 was found to be associated with the apical and basolateral solutions, while no detectable amount of [^3^H]OligoGM1 was found within the cells (Figure 4). This suggests that hBLECs do not internalize OligoGM1, again leading to speculation about a passive paracellular route for OligoGM1.

### 2.4. Metabolic Integrity of hBLEC-Crossed OligoGM1 

To understand if the radioactivity found in the basolateral side after the hBLEC transport experiment indeed corresponded to intact OligoGM1, we verified its metabolic stability by high-performance thin layer chromatography (HPTLC) autoradiography using a Beta IMAGER™ Tracer system (Biospace Lab) with the M3 Vision software (M3Vision Analysis Software, Biospace Instruments, Paris, France). As shown in Figure 5, metabolically stable [^3^H]OligoGM1 was found in the receiving compartment, suggesting that OligoGM1 is not metabolized by hBLECs. Indeed, no other radioactive compounds were identified, considering that the detection limit of the instrument is 15 dpm. This result finally proves that the radioactivity found in the receiving compartment does not correspond to [^3^H]galactose or to any other possible new glycan, but only to intact original penta-saccharide.

### 2.5. Characterization of OligoGM1 Transport across the hBLEC Monolayer

#### 2.5.1. Time-Dependent Transport

To find out if a time dependence existed, the OligoGM1 (50 μM plus 1 × 10^6^ dpm) transport from apical to basolateral (A → B) directions was studied up to 240 min. As shown in Figure 6a, the OligoGM1 quantity increased progressively over time in the basolateral compartment, thus demonstrating a linear time dependence for the OligoGM1 transport within the time range analyzed.

#### 2.5.2. Concentration-Dependent Transport

To demonstrate the possibility of a dose dependence in transport, we performed the apical to basolateral (A → B) transport experiments at various OligoGM1 concentrations (5 to 300 μM plus 1 × 10^6^ dpm). We pointed out that the OligoGM1 quantity recorded in the basolateral compartment after 60 min presents a proportional increase with respect to the considered concentrations (Figure 6b).

#### 2.5.3. Influence of BSA

In order to evaluate the possible interactions with molecules that could influence OligoGM1 availability to hBLECs and the transport to the receiving side, 0.1% (*w*/*v*) bovine serum albumin (BSA) was added to 50 µM plus 1 × 10^6^ dpm OligoGM1-RH solutions to mimic the blood serum protein content [35]. No difference in the rate of transport was revealed after 60 min of incubation by the comparative analysis of BSA-OligoGM1 and OligoGM1 solutions—excluding a BSA disturbance on the OligoGM1 passage—in the tested conditions (Figure 6c), suggesting the possibility of no interference of serum protein in the availability of OligoGM1 in vivo.

#### 2.5.4. Transport at 4 °C 

A direct transport experiment (A → B) for OligoGM1 was performed at 4 °C in order to inhibit the active component of the transport across the BBB endothelial monolayer [36,37]. OligoGM1 Papp measured at 4 °C was slightly reduced with respect to the transport at 37 °C (Figure 6d). Additionally, as shown in the graph, LY Papp, which is a marker of passive transport through hBLECs [28], presents a slight reduction at 4 °C. These data are in accordance with the law for the passive transport (Equation (4) in the Appendix A), for which the Papp at 4 °C decreases slightly but not negligibly, confirming the passive route for LY [28,38,39] and suggesting the same transport behavior for OligoGM1.

A further suggestion that OligoGM1 moves passively across the hBLEC barrier comes from Equation (5) in the Appendix A, which assumes that two different molecules (i.e., OligoGM1 and LY) that move passively from one compartment to another at a given temperature can be compared by considering the ratio between their Papp and the ratio of their respective hydrodynamic radius (r).

Thus, it is possible to directly compare the permeability of different molecules as long as their hydrodynamic radius is known. The hydrodynamic radius of LY is reported to be 4.9 Å [40], while the OligoGM1 hydrodynamic radius has not been determined yet. Still, we can make a reasonable estimate from literature data on analogous species, such as the linear tetrasaccharide stachyose (6.5 Å) [41] and the branched pentasaccharide 6′-R-maltosyl-maltotrioside (8.7 Å) [42]. Using these values, we can consider the hydrodynamic radius of the OligoGM1 to lie between 6.5 and 8.7 Å. Thus, the ratio of r LY to r OligoGM1 is determined as 0.56 < x < 0.75 (Figure 6d).

At 37 °C, the ratio between Oligo Papp (0.88 ± 0.024) and LY Papp (1.27 ± 0.072) turns out to be 0.69, while at 4 °C the ratio between OligoGM1 Papp (0.67 ± 0.017) and LY Papp (1.04 ± 0.068) is equal to 0.64 (Figure 6d).

It follows that for LY and OligoGM1, the relationship between the Papp at a given temperature and the inverse of the respective hydrodynamic radius values are in great agreement (Figure 6d), supporting the hypothesis that the OligoGM1 moves using the same transport mechanism as LY, which is reported to be passive transport [28,34].

### 2.6. Interaction with Transporters or Efflux Pumps

Bidirectional transport analysis represents a common indirect strategy to evaluate the implication of active transport in drug absorption [36,43,44,45,46,47]. For this purpose, many indices and coefficients can be calculated and compared with other data to characterize the transport of a drug across an epithelial-type barrier [32,46,48,49,50,51]. Thus, OligoGM1 was administered to the hBLECs in the basolateral compartment or apical compartment of the transwell system. In order to determine the OligoGM1 efflux ratio (E) and make a comparison with existing data about drugs, the ratio between Papp in relation to the transport in both the directions, i.e., from apical to basolateral (A → B) and from basolateral to apical (B → A), was calculated according to Equation (6) as reported in Appendix A (Figure 7). The resulting E values were equal to 0.99 for OligoGM1 and 1.25 for LY. Published data state that an efflux ratio < 2 is an indicator of the passive paracellular transport implication [52,53,54].

In order to directly exclude the implication of the active mechanism in the OligoGM1 transport, Caco2-cells were employed as a naturally overexpressing ABC transporter cellular model to test a possible interaction of OligoGM1 with the ABC protein family, according to the experimental approach recently reported by Sevin et al. [29]. Following evaluation of Caco-2 cell viability upon OligoGM1 treatment (Appendix A), the OligoGM1 interaction with P-gp/BCRP and with MRP was assessed.

As shown in Figure 8a,b, verapamil and MK571, specific competitor inhibitors of P-gp/BCRP and MRPs [55,56], significantly decrease the outgoing rate (Kout) of R123 and CMFDA by more than 30%, showing that the system was properly working (Figure 8). Importantly the fluorescent signal-revealing analysis applied for the detection of R123 and CMFDA did not identify any difference in the efflux of the substances from the transporters in concomitance to OligoGM1 treatments at the tested concentrations with respect to the control condition (Figure 8). These data agree with the above indirect observation and contribute to excluding the implication of the ABC-mediated active mechanism in OligoGM1 transport.

### 2.7. Maintenance of Neurotrophic Properties for hBLEC-Crossed OligoGM1 

To further verify if OligoGM1 maintains its bioactive properties after crossing the BBB model, we evaluated the hBLEC-crossed OligoGM1 neurodifferentiative potential on N2a cells, as previously reported [22,23]. Thus, we performed an apical to basolateral transport experiment (A → B), employing 300 μM OligoGM1 applied to the upper compartment. After 240 min, we collected the receiving compartment solution, which contained 25% of the OligoGM1 initially administered to the apical side, corresponding to ~25 μM OligoGM1, as expected from time–concentration-dependent experiments reported above (Figure 6a,b). Subsequently, N2a cells were incubated with the basolateral solution containing the hBLEC-crossed OligoGM1. As shown in Figure 9a, hBLEC-crossed OligoGM1 induced a neuron-like morphology after 24 h. In accordance with the OligoGM1 behavior reported in previous studies [22,23], the N2a receiving hBLEC-crossed OligoGM1 showed a clear neurite sprouting that was instead absent in the round-shaped control cells.

Additionally, we previously reported that OligoGM1 induces N2a neuritogenesis by eliciting the activation of the TrkA-ERK1/2 pathway [22,23]. Importantly here, by immunoblotting, we highlighted the activation of the TrkA-ERK1/2 pathway by hBLEC-crossed OligoGM1, as shown in Figure 9b. These results further confirm that OligoGM1 maintains its neurotrophic potential after crossing the BBB human model.

## 3. Discussion

In 1976, a mix of gangliosides extracted from calf brains was approved as a therapeutic agent by the name of Cronassial for the treatment of peripheral neuropathies [57,58,59,60]. Subsequently, a drug composed of purified GM1 entered the pharmaceutical market for the treatment of neurodegenerative diseases and damage to the CNS and spinal cord [61]. Several clinical trials were performed with peripheral or intraventricular brain administration of GM1. Important clinical success has been achieved in the treatment of peripheral neuropathology with GM1 [62,63,64,65] and in ischemic brain injuries [66,67], conditions in which there is no BBB to cross or it is severely damaged.

In neurodegenerative diseases such as Parkinson’s and Alzheimer’s diseases, despite the great preclinical successes [18,19,68,69,70,71], GM1 did not reach a sufficient amount in the brain target site to perform its reparative functions. Probably the reason is that these clinical trials included peripheral administration of the ganglioside, which inefficiently permeates through the BBB. Although the BBB has been reported to be dysfunctional [12,13,72,73,74,75,76], it remains a huge obstacle to adequate drug delivery to the brain.

The investigation of GM1 as a drug for clinical treatment slowed down in the 1990s, following the hypothesis that circulating gangliosides could induce the formation of auto-antibodies responsible for Guillain-Barré syndrome [77,78]. This hypothesis was subsequently disproved—gangliosides, and therefore also GM1, alone are not immunogenic [79]. To date, although the missing association between GM1 injection and Guillain-Barré syndrome development seems to have been accepted, the disadvantage of GM1′s inability to cross the BBB still means its potential is underexploited.

Importantly, our studies to disclose the mechanism of action underlying the neurotrophic and reparative properties of GM1 have revealed that the bioactive portion of the molecule is the oligosaccharide chain [22,23,24,26,27], which surprisingly, if administered peripherally into mice, is detectable within the brain [25]. Even more noteworthy is the finding that OligoGM1 injection in a sporadic Parkinson’s disease model, the *B4galnt1^+/-^* mouse, in which the neuronal GM1 content is decreased due to a reduced expression of *B4galnt1* glycosyltransferase, promoted a recovery of both behavioral and biochemical features, reaching healthy conditions [25].

On this premise, the soluble and hydrophilic oligosaccharide chain of GM1 (Figure 1) seems to be the long-awaited candidate to replace GM1 use in the clinic for the treatment of neurodegenerative diseases. To propose OligoGM1 as a new drug able to act on the CNS, it is essential to demonstrate its ability to penetrate the brain by crossing the BBB in order to reach the target neurons. Thus, in the present study we present details of the OligoGM1 transport across the BBB by using a human in vitro BBB model.

Accordingly, we took advantage of a human BBB model derived from stem cells involving a co-culture of CD34^+^-derived endothelial cells and brain pericytes in a transwell system [28]. The CD34^+^-derived endothelial cells are initially co-cultured with brain pericytes to induce BBB properties. After 6 days of co-culture, the resulting hBLECs express TJ proteins at cell–cell contacts and transporters typically observed in the brain endothelium, maintaining the properties of in vivo BBB for at least 20 days [28], a period during which we performed transport experiments. This model is now widely used to investigate BBB physiology, as well as molecules and cells passage across the BBB [80,81,82,83,84,85]. The loading of the human BBB model with GM1 or OligoGM1, followed by the measure of permeability parameters, provided evidence to consider OligoGM1 as a molecule transported by the paracellular mechanism, and importantly with a rate 20-fold higher then ganglioside GM1 (Figure 3). Specifically, the endothelial permeability of OligoGM1 and Lucifer yellow, here used as a paracellular route marker [28,34], highlighted comparable values, which suggests the possible use of a paracellular passage route by OligoGM1 also (Figure 2 and Appendix A).

Accordingly, we found that hBLECs do not internalize (Figure 4) nor metabolize the OligoGM1 (Figure 5), which indeed remains intact and biologically active. In fact, after crossing by hBLECs, OligoGM1 was able to induce neuronal differentiation by activating the TrkA-ERK1/2 pathway in neuroblastoma N2a cells (Figure 9), thus maintaining its reported neuronal properties [22,23,24]. The 4 °C direct transport experiments provide more evidence of the implications of a passive route, since Papp depends on the absolute temperature, as shown in Equations (1)–(4) in the Appendix A. The differences found between Papp at 4 and 37 °C for OligoGM1 (Figure 6d), as well as for LY, are in agreement with the expected ones for passive diffusion.

A further confirmation comes from the bidirectional experiments, which resulted in an OligoGM1 minor efflux ratio of 2 (Figure 7), which is an indicator of passive paracellular passage. as reported [52,53,54].

Moreover, we directly excluded the possible involvement of ABC transporters (Figure 8), P-glycoprotein (P-gp), breast cancer resistance protein (BCRP) and the multidrug-resistance-associated proteins (MRPs), using the recently described “pump out” system based on the use of Caco-2 cells [29].

Finally, we found that the OligoGM1 passage across the human BBB model is time- and concentration-dependent, but it was not influenced by the presence of BSA, suggesting the possibility of no interference of serum protein on the availability of OligoGM1 in vivo (Figure 6).

In conclusion, our present findings demonstrate that the soluble oligosaccharide portion of GM1 is able to cross the human BBB model more efficiently than GM1, maintaining its metabolic integrity and biological activity [55,56]. The hBLEC transport experiments, together with the experimental data that exclude the involvement of the active ABC transporters P-gp, BCRP, and MRPs, allow us to state with a high degree of confidence that OligoGM1 crosses the BBB through a passive paracellular transport, while the active transcellular transport could be reasonably excluded.

Our work provides exciting evidence for the use of OligoGM1 in clinical settings for the treatment of neurodegenerative diseases for which a positive response with GM1 has been demonstrated, firstly for the treatment of sporadic Parkinson’s disease. 

## 4. Materials and Methods 

### 4.1. Materials

The chemicals purchased were of the highest available purity, the solvents were distilled before use, and the water was doubly distilled using glass instruments.

Cell culture plates, Transfectagro™ reduced serum medium (Transfectagro), and Matrigel^®^ growth-factor-reduced and transwell tissue culture insert (1.12 cm^2^, polycarbonate membrane, 0.4 μm pore size) were from Corning (Corning, Corning, NY, USA). Sciencell™ endothelial cell medium (ECM) and endothelial cell growth supplement (EGCS) were from Sciencell Research Laboratories (Sciencell, Carlsbad, CA, USA). Mouse neuroblastoma Neuro2a (N2a) cells (RRID: CVCL_0470), calcium- and magnesium-free phosphate-buffered saline (CMF-PBS), ethylenediaminetetraacetic acid (EDTA), d-mannitol, trypsin, pig skin gelatin type A, LY, BSA, 2-propanol, formic acid, MTT, mouse α-tubulin (α-tub) antibody (RRID:AB_477579), rhodamine 123 (R123), MK571, verapamil, sodium orthovanadate (Na_3_VO_4_), phenylmethanesulfonyl fluoride (PMSF), aprotinin, and protease inhibitor cocktail (IP) were from Sigma-Aldrich (Sigma-Aldrich, St. Louis, MO, USA). Dulbecco’s modified Eagle’s medium (DMEM), high-glucose DMEM, fetal bovine serum (FBS), fetal calf serum (FCS), L-glutamine, non-essential amino acids without L-glutamine, penicillin/streptomycin solution, and RH buffer were from EuroClone (Euroclone, Paignton, UK). Trypsin-EDTA and gentamycin were from BiochromAG (Berlin, D). Anti-mouse IgG (H+L) antibody (RRID: AB_228307) and green CMFDA were from Thermo Fischer Scientific (Thermo Fischer Scientific, Waltham, MA, USA). Rabbit anti-TrkA (RRID: AB_10695253), rabbit anti-phospho-TrkA (tyrosine 490, Tyr490) (RRID: AB_10235585), rabbit anti-p44/42 MAPK (ERK1/2) (RRID: AB_390779), rabbit anti-phospho-p44/42 MAPK (pERK1/2) (Thr202/Tyr204) (RRID:AB_2315112), and anti-rabbit IgG (RRID: AB_2099233) antibodies were from Cell Signaling Technology (Cell Signaling Technology, Danvers, MA, USA). The chemiluminescent kit for Western blot was from Cyanagen (Cyanagen, Bologna, Italy). Ultima Gold liquid scintillation cocktail was from Perkin Elmer (Perkin Elmer, Waltham, MA, USA). HPTLC and Triton X-100 were from Merk Millipore (Merk Millipore, Frankfurt, Germany). The 4–20% Mini-PROTEAN^®^ TGX™ precast protein gels, turbo polyvinylidene difluoride (PVDF), mini-midi membrane, and DC™ protein assay kit were from BioRad (BioRad, Hercules, CA, USA).

Bovine pericytes and CD34^+^-derived human endothelial cells (CD34^+^-ECs) were obtained from the Stable Human BBB kit™ (LBHE, Blood–brain Barrier Laboratory, Artois University, Arras, Cedex, France). Briefly, this model reproduces the BBB by cultivating CD34^+^-derived endothelial cells on a Matrigel™-coated insert in the presence of brain pericytes. Human umbilical cord blood was used to extract and purify the CD34^+^ cells, which were subsequently differentiated into endothelial cells, as previously described [86]. All the experiments were done with the authorization of French Ministry of Higher Education and Research (CODECOH DC2011-1321, January 2013) for the collection of human cells and after the infants’ parents signed consent form.

Caco-2 cells were from Sigma-Aldrich (Lyon, France) and were cultivated as previously described to quickly and efficiently assess the functionality of P-gp/BCRP and MRPs [29].

### 4.2. Ganglioside GM1 and Oligosaccharide Preparation

Ganglioside GM1 (II^3^Neu5Ac-Gg_4_Cer, β-Gal-(1-3)-β-GalNAc-(1-4)-[α-Neu5Ac-(2-3)]-β-Gal-(1-4)-β-Glc-(1-1)-Cer) was purified from the sialidase-treated ganglioside mixture extracted according to the tetrahydrofurane/buffer procedure [87] from fresh pig brains provided by the slaughterhouse of the Galbani Company, Melzo, Italy. Ganglioside nomenclature is in accordance with IUPAC-IUBB recommendations [88]. The ganglioside mixture (5 g as sialic acid) was dissolved in 500 mL of prewarmed (36 °C) 0.05 M sodium acetate, 1 mM CaCl_2_ buffer, at pH 5.5. *Vibrio cholerae* sialidase (1 unit) was added to the solution every 12 h [89]. Incubation at 36 °C and under magnetic stirring was maintained for two days, and the solution was dialyzed at 23 °C for 4 days against 10 L of water, which was changed 5 times a day. The sialidase-treated ganglioside mixture was dried, solubilized in methanol, and applied to a 50 × 2 cm diethylaminoethyl (DEAE) column in methanol, according to the overloading procedure [90]. The eluted GM1 fractions, identified by TLC, were pooled, dried, and submitted to a 150 × 2 cm silica gel column for chromatography using the chloroform/methanol/water solvent system at 60:35:5 by volume. Fractions containing pure GM1 were collected and dried. The residue was dissolved in chloroform/methanol at 2:1 by volume and precipitated by adding 4 volumes of cold acetone. After centrifugation (15,000× *g*), the GM1 pellet was separated from the acetone, dried, dissolved in 50 mL of deionized water, and lyophilized, giving 1350 mg of white powder, which was stored at −20 °C.

GM1-containing tritium at position 6 of external galactose was prepared by enzymatic oxidation with galactose oxidase, followed by reduction with sodium boro[^3^ H]hydride [91].

The oligosaccharides OligoGM1 (II^3^Neu5Ac-Gg_4_, β-Gal-(1-3)-β-GalNAc-(1-4)-[α-Neu5Ac-(2-3)]-β-Gal-(1-4)-β-Glc), and [^3^H]OligoGM1 (II^3^Neu5Ac-[^3^H]Gg_4_, [6-^3^H]β-Gal-(1-3)-β-GalNAc-(1-4)-[α-Neu5Ac-(2-3)]-β-Gal-(1-4)-β-Glc-Cer) were prepared with minor changes from the old procedure using ozonolysis, followed by alkaline degradation [92] of GM1 and [^3^H]GM1, respectively. Briefly, GM1 or [^3^H]GM1 was dissolved in methanol and slowly saturated with ozone at 23 °C. The solvent was then evaporated under vacuum and the residue was immediately brought to pH 10.5–11.0 by addition of triethylamine. After solvent evaporation, OligoGM1 and [^3^H]OligoGM1 were purified by flash chromatography using chloroform/methanol/2-propanol/water at 60:35:5:5 by volume as the eluent. OligoGM1 and [^3^H]OligoGM1 were dissolved in methanol and stored at 4 °C.

Altogether, nuclear magnetic resonance (NMR), mass spectrometry (MS), high-performance thin layer chromatography (HPTLC) (HPTLC), and autoradiographic analyses showed a homogeneity of over 99% for OligoGM1, as shown in Appendix A [24].

### 4.3. Cell Cultures

#### 4.3.1. Bovine Pericytes

Bovine pericytes (1 × 10^6^ cells) were extracted, purified, cultured, and frozen in liquid nitrogen, as previously described [93]. Before use, frozen vials were immediately thawed in a water bath at 37 °C. Cell suspension was kindly homogenized using an automatic pipette and spread on a 100 mm diameter Petri dish previously coated with 0.2% pig skin gelatin (*w*/*v*, in CMF-PBS), 0.2 g/L KH_2_PO_4_, 8.0 g/L NaCl, 2.87 g/L, Na_2_HPO_4_-12H_2_O, and 0.2 g/L KCl), with pre-warmed complete medium for human BBB (ECM medium supplemented with 1% EGCS, 5% FCS, and 50 μg/mL gentamycin). After 3 h, cells were observed to ensure their attachment. Thus, the medium was replaced with a fresh one to eliminate the toxic DMSO residue. Pericytes were grown at 37 °C in a humidify atmosphere of 95% air/5% CO_2_ for two days.

#### 4.3.2. CD34^+^-ECs

Endothelial cells (1 × 10^6^ cells) derived from umbilical cord blood hematopoietic stem cells (CD34^+^-ECs) underwent very fast thawing at 37 °C in a water bath. Subsequently, cell solution was quickly and kindly suspended by an automatic pipette and plated in a 100 mm diameter Petri dish that had been previously coated with 0.2% pig skin gelatin (*w*/*v*, in CMF-PBS), with pre-warmed complete medium for human BBB (ECM medium supplemented with 1% EGCS, 5% FCS, and 50 μg/mL gentamycin). Cells were incubated for 3 h to ensure their attachment and subsequently the medium was changed to prevent unwanted cell contact with toxic DMSO. Endothelial cells were grown at 37 °C in a humidify atmosphere of 95% air/5% CO_2_ for two days.

#### 4.3.3. Co-Culture of CD34^+^-ECs and Bovine Pericytes

For transport experiments, the human in vitro co-culture BBB model was prepared as previously described [28]. Briefly, pericytes were sub-cultured after two days of culture, when cells reached confluency. Cells were detached by trypsin-EDTA solution (0.05–0.02%, *w*/*v* in CMF-PBS) and 5 × 10^4^ pericytes were seeded into each well of 12-well plates pre-coated with gelatin (0.2% pig skin gelatin *w*/*v*, in CMF-PBS) in complete BBB medium and incubated at 37 °C in a humidify atmosphere of 95% air/5% CO_2_. After 3 h from the point of plating pericytes, CD34^+^-ECs were trypsinized, seeded at a density of 8 × 10^4^/insert onto the Matrigel-coated Transwell inserts (1.12 cm^2^, polycarbonate membrane, 0.4 μm pore size, Corning), and cultured in BBB complete medium. Matrigel used for coating was diluted 48 times in DMEM and rinsed once with DMEM before seeding cells.

To induce BBB properties, transwell-seeded CD34^+^-ECs were co-cultured with pericytes, placing the inserts on the pericyte-filled 12-wells (Appendix A) and incubating them at 37 °C in a humidify atmosphere of 95% air/5% CO_2_. After 6 days of co-culture, CD34^+^-ECs were completely differentiated into hBLECs and could be used for transport experiments for at least 20 days (Appendix A). As a background for permeability tests, transwell inserts covered and filled with Matrigel and medium but without cells were used. Medium was completely renewed every 2 days.

#### 4.3.4. Caco-2 Cells

Caco-2 cells were seeded as previously reported [29]. Briefly, 0.4 × 10^5^ Caco-2 cells were seeded on 25 cm^2^ plastic flasks and changed every second day with DMEM high-glucose medium with L-glutamine (584 mg/L), supplemented with 10% of FBS/FCS (*v*/*v*), 1% of non-essential amino acids without L-glutamine (*v*/*v*), and 1% of penicillin/streptomycin solution (*v*/*v*). Caco-2 cells were trypsinized after 3 days of incubation while they covered 80–90% of the flask and seeded at a density of 5 × 10^5^ in 75 cm^2^ flasks in complete medium. After 5 to 6 days, Caco-2 cells reached high cell density (>5 × 10^4^ cells/cm^2^) and were then split into rat tail collagen type I 96-well plates at 30 × 10^3^ cells/cm^2^ and cultivated in the complete medium for 6 days. Medium was replaced every second day. After a cell culture period of 6 days, Caco-2 cells were used for transport experiments. Cells were maintained at 37 °C in a humidified atmosphere of 95% air/5% CO_2_.

#### 4.3.5. N2a Cells

N2a cells were cultured and propagated up to the 25th passage in DMEM high-glucose medium supplemented with 10% heat-inactivated FBS (*v*/*v*), 1% L-glutamine (*v*/*v*), and 1% penicillin/streptomycin (*v*/*v*) at 37 °C in a humidified atmosphere of 95% air/5% CO_2_. Cells were sub-cultured twice a week at the 80–90% confluence, as previously reported [22,23,24].

For differentiation experiments, N2a cells were seeded at 5 × 10^3^/cm^2^ and incubated for 24 h in regular culture condition to allow cell attachment. Before all treatments, growth medium was removed and cells were adapted for 30 min in pre-warmed (37 °C) Transfectagro medium containing 2% heat-inactivated FBS, 1% L-glutamine, and 1% penicillin/streptomycin solution. All treatments were conducted at 37 °C in a humidified atmosphere of 95% air/5% CO_2_.

### 4.4. Endothelial Permeability (Pe) Measurement

Prior to the experiments, in order to verify the hBLECs’ monolayer integrity and stability, the permeability parameters were verified as previously reported [28]. Briefly, 1.5 mL Ringer–Hepes solution (RH) (150 mM NaCl, 5.2 mM KCl, 2.2 mM CaCl_2_, 0.2 mM MgCl_2_6H_2_O, 6 mM NaHCO_3_, 5 mM HEPES, pH: 7.4) was added to empty wells of a 12-well plate. Filter insert containing confluent monolayers of hBLECs or not (control) were subsequently placed in the 12-well plate, filled with 0.5 mL RH buffer containing the fluorescent integrity marker LY (50 μM), and then placed on an orbital shaker (150 rpm, Heidolph Titramax 100) at 37 °C in a humidify atmosphere of 95% air/5% CO_2_. Every 20 min for 1 h, filter inserts were withdrawn from the receiver compartment and placed in the next one. To verify the quantity of LY, at the end of the experiments, aliquots from the donor solution were taken at the end of the experiments and the fluorescence was quantified using a quantification curve of LY as referee. At least three inserts with cells and three without cells were tested in each permeability measurement. The fluorescence was measured using a luminescence spectrometer (LS 50B, Perkin Elmer), using the following excitation/emission wavelength (nm) settings: 432/538. Pe parameters were calculated according to the formulas shown in the Appendix A (Equations (7)–(10)). In parallel, as positive control for breakdown of hBLECs, d-mannitol, was used at a concentration of 1.4 M. At this concentration, mannitol induces an osmotic shock, and thus an increase in paracellular permeability [94,95,96].

### 4.5. Evaluation of Toxicity of GM1 and OligoGM1 on hBLECs

To evaluate the toxicity of GM1 and OligoGM1, the LY endothelial permeability (Pe) and apparent permeability (Papp) were considered for evaluation of hBLEC monolayer integrity after administration of compounds. The increase in LY Pe or Papp in the presence of an added substance with respect to the Pe or Papp of LY alone should suggest an alteration in the TJ apparatus due to a cytotoxic effect exerted by the added substance.

To examine the LY permeability, 1.5 mL of RH buffer was added to empty wells of a 12-well plate, then hBLEC filters were placed in the 12-well plate and filled with 0.5 mL RH buffer containing 50 μΜ LY plus OligoGM1 or GM1 at an appropriate test concentration (from 5 to 300 μΜ). The hBLEC inserts were moved every 20 min to a new RH-filled well up to 60 min. At the end of the incubation period, solutions were collected from apical and basolateral compartments, LY was quantified, and Papp was calculated as reported in the Appendix A (Equation (1)) and previously mentioned [28].

### 4.6. Fate of OligoGM1 and GM1 in hBLECs

The possible internalization of OligoGM1 in hBLECs was determined by measuring the radioactivity associated to the cells. Briefly, at the end of direct transport, apical and basolateral solutions were collected, hBLEC filters were washed with RH buffer, and cells were detached by trypsin and lysed with 1mM Na_3_VO_4_, 1 mM PMSF, 2% (*v*/*v*) aprotinin, and 1% (*v*/*v*) IP in cold PBS. At this point, the lysed cell solution was counted to measure the radioactivity. All the cell lysates were combined with 5 mL of ULTIMA GOLD liquid (PerkinElmer), shaken, and counted for 20 min by liquid scintillation analyzer (TRI-CARB 2100TR, Packard). The resulting dpm radioactivity measures were used to estimate the molecule quantity.

### 4.7. Evaluation of OligoGM1 Metabolic Integrity after hBLEC Transport

Since OligoGM1 is tritium labeled at position 6 of the external galactose, any possible galactosidase activity present on the plasma membrane of hBLECs can result in external galactose removal, leading to loss of radioactivity from the penta-saccharide. Radiolabeled galactose may pass the BBB or may enter into the cells and be recycled for the biosynthesis of new glycans, so that the radioactivity found may correspond to the galactose or to any possible new glycan rather than to the original oligosaccharide.

Thus, to test the integrity of the OligoGM1 molecule after hBLEC crossing, OligoGM1 solution was prepared in RH buffer at a final concentration of 100 μM plus 1 × 10^6^ dpm of tritium-labeled OligoGM1. For the apical–basolateral transport experiment, the hBLEC insert’s apical side was filled with 0.5 mL of the drug solution and placed on 1.5 mL RH buffer-filled well of a 12-well plate. After 60 min, the solution of the receiving compartment (about 1.5 mL) was collected and 200 μL were combined with 5 mL of ULTIMA GOLD liquid (PerkinElmer), shaken, and counted for 20 min by a liquid scintillation analyzer (TRI-CARB 2100TR, Packard).

To verify that the counted radioactivity indeed corresponded to [^3^H]OligoGM1, 0.5 mL of receiving compartment solution was lyophilized and suspended in 0.5 mL of methanol. Then, 50 μL of the methanol solution (1/10 of the starting material) was separated by HPTLC using the chloroform/methanol/0.2% CaCl_2_ solvent system at 60:35:8 by volume. The [^3^H]OligoGM1 was visualized with a BetaIMAGER™ tracer system (Biospace Lab) using an acquisition time of 17 h, and was identified using pure [^3^H]OligoGM1 as standard.

### 4.8. Transport Experiments

Transport experiments were carried out by adding tritium-labeled GM1 or OligoGM1 (1 × 10^6^ dpm) to cold GM1 or OligoGM1, respectively, at different concentrations, as specified below.

To identify and quantify the GM1 or OligoGM1 molecules in every transport experiment, aliquots from apical A (5 μL) and basolateral B (200 μL) compartments were collected at indicated time points and combined with 5 mL of ULTIMA GOLD liquid (PerkinElmer), shaken, and counted for 20 min by liquid scintillation analyzer (TRI-CARB 2100TR, Packard). Resulting radioactivity measures were used to estimate the quantity of the molecules. Experiments were performed on an agitation plate at 37 °C (150 rpm, Heidolph Titramax 100) in a humidify atmosphere of 95% air/5% CO_2_. At least three inserts with cells were tested in each transport experiment for each condition.

#### 4.8.1. Direct Transport (Apical to Basolateral A → B)

To examine flux from apical to basolateral compartments (A → B), 1.5 mL RH buffer was added to empty wells of a 12-well plate. Following this, the hBLEC filter was placed in the same 12-well plate and filled with 0.5 mL RH buffer containing 1 × 10^6^ dpm of tritiated [^3^H]GM1 or [^3^H]OligoGM1 plus 50 μM of cold GM1 or OligoGM1 (Appendix A).

To establish the time dependence of the transport experiments, hBLEC inserts filled with OligoGM1 were transferred into a new receiver well plate filled with fresh RH buffer every 20 min up to 240 min. The hBLECs’ properties, integrity, and stability were maintained up to 240 min in the transport experiments with RH buffer in the upper and basolateral side (see [28] and Appendix A).

To establish the dose dependence in the transport experiments, 1 × 10^6^ dpm of tritiated [^3^H]OligoGM1 plus 5, 10, 50, 100, and 300 μM OligoGM1 solutions were prepared and employed as described above. The hBLECs inserts were transferred into a new RH-filled receiver well plate every 20 min up to 60 min.

To evaluate the possible influence of plasma serum on OligoGM1 transport, RH buffer + 0.1% BSA (*w*/*v*) was used as the working solution for the apical side to mimic the protein content of plasma serum [35]. The experiments were conducted for up to 60 min.

To evaluate the possible active transport involvement, the transport of OligoGM1 from apical to basolateral (A → B) compartments was performed at 4 °C, with the aim of inhibiting the active transport mechanism at low temperature.

#### 4.8.2. Indirect Transport (Basolateral to Apical B → A)

For the inverse transport study (related to basolateral → apical flux, B → A), 1.5 mL of 1 × 10^6^ dpm of tritiated [^3^H]OligoGM1 plus 50 μM OligoGM1 in RH buffer were added to an empty well of a 12-well plate (Appendix A). Every 20 min up to 60 min, the hBLEC filter filled with 0.5 mL RH buffer was removed and replaced with a new filter with fresh RH buffer.

### 4.9. Establishment of Permeability Parameters

The apparent permeability, efflux ratio, and endothelial permeability were calculated to estimate the absorption processes of drugs [97,98,99,100,101,102], to predict implication of active transport in the process of drugs absorption [36,43,44,45,46,47,48,52,53,54], and to more accurately appreciate the transport of a drug across an endothelial barrier [103,104], respectively. Detailed information on the equations used to calculate the permeability parameters are reported in the Appendix A.

### 4.10. ABC Transporter Competition Assay

The Caco-2 “pump out” assay was performed as previously reported [29]. Briefly, Caco-2 cells in 96-well plates were washed once with RH buffer and incubated for 120 min with 10 µM R123 or for 15 min with 0.5 µM CMFDA. R123 and CMFD are substrates for the major efflux pumps expressed by BBB and intestine cells, i.e., P-gp/BCRP and MRPs, respectively. After the incubation period, Caco-2 cells were washed twice with RH buffer with or without test compounds (OligoGM1 at 10, 50, and 100 µM). The rate out (Kout) of Caco-2 cells for fluorescent dyes was monitored by fluorescence measurement (λex = 501 nm and λem = 538 nm for R123; λex = 485 nm and λem = 538 nm for CMFDA) using a microplate fluorescent reader (BioTek, H1, Vermont, Winooski). The amount of dye expelled from the cells was measured at 37 °C and every 5 min for one hour for R123 and every 2 min for 30 min for CMFDA. The rate out of the dye from cells was calculated as the slope of the curve of the cumulative amount of dye against the time (Kout = ∆pmol⁄∆t) and compared with the value in the presence of test compounds. Verapamil and MK571 substrates and competitive transport inhibitors of P-gp/BCRP [56] and MRPs [55], respectively, were used as positive controls for efflux pumps activity. Evaluation of OligoGM1 toxicity on Caco-2 cells was performed by MTT assay (for procedural details, see Appendix A).

### 4.11. Evaluation of OligoGM1′s Functional Stability after hBLEC Transport

In order to demonstrate the maintenance of OligoGM1 biological activity after crossing the hBLEC monolayer, we tested the hBLEC-crossed OligoGM1 ability to induce N2a differentiation throughout the TrkA-ERK1/2 pathway activation as shown previously [22,23,24] by using a basolateral OligoGM1-receiving solution. Precisely, [^3^H]OligoGM1 (1 × 10^6^ dpm) diluted into cold OligoGM1 solution (300 µM) was administered for 240 min to hBLECs in the upper compartment so that at a volume of 0.5 mL there was 150 nmol of OligoGM1. In this case, the apical and basolateral receiver compartments were filled with Transfectagro medium containing 2% of FBS and tested to ensure they did not alter the hBLEC characteristics by measuring LY permeability (Appendix A). After the 240 min incubation period, the basolateral solution was screened to find the quantity of crossed OligoGM1 by measuring ^3^H-radioactivity and was then administered to N2a cells for 24 h. For control cells, we administered a basolateral solution derived from hBLECs incubated with Transfectagro without OligoGM1. Morphological signals of neurodifferentiation were evaluated by cell observation and TrkA-ERK1/2 pathway activation was verified by immunoblotting (for procedural details see Appendix A and [22,23]).

### 4.12. Protein Determination

Protein concentrations of samples were assessed using a DC™ protein assay kit according to the manufacturer’s instructions, using BSA as standard.

### 4.13. Statistical Analysis

Data are expressed as mean ± SEM. When the normality of data was not assessed because the number of the samples was too small, data were analyzed for significance by Mann–Whitney test. Otherwise, two-way ANOVA test was applied. The analysis was performed with Prism software (GraphPad Software, version 8.0, Inc. La Jolla, CA, USA).

### 4.14. Other Analytical Methods

NMR spectra were recorded with a Bruker AVANCE-500 spectrometer at a sample temperature of 298 K. NMR spectra were recorded in CDCl3 or CD3OD and calibrated using the TMS signal as internal reference [22,24]. Mass spectrometric analysis was performed in positive electrospray ionization mass spectometry (ESI-MS) mode. MS spectra were recorded on a Thermo Quest Finningan LCQTM DECA ion trap mass spectrometer equipped with a Finnigan ESI interface; data were processed by a Finnigan Xcalibur software system. All reactions were monitored by TLC on silica gel 60-well plates (shown in Appendix A; also see [22,24].

No randomization or blinding procedures were performed for our experiments. Custom-made materials will be shared upon reasonable request. The *n* number (*n* =) in the figure legend indicates the number of independent cell culture preparations.

## Figures and Tables

**Figure 1 ijms-21-02858-f001:**
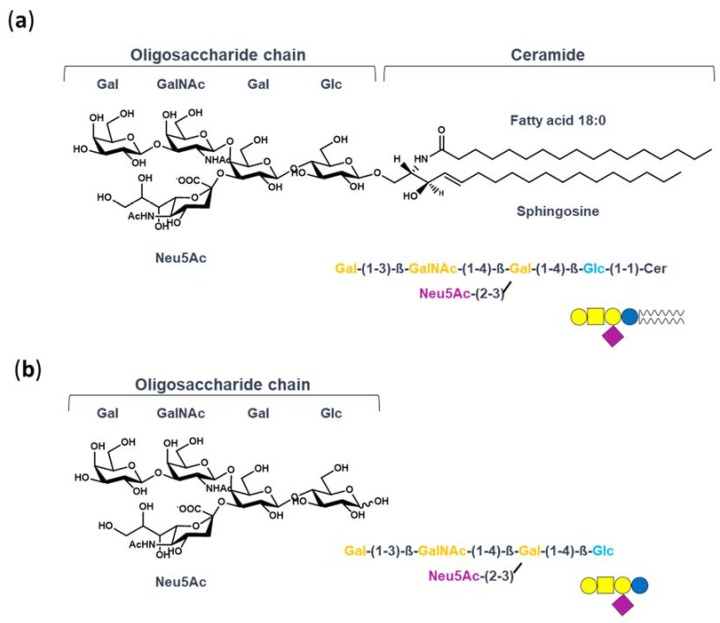
Structure of GM1 and OligoGM1: (**a**) ganglioside GM1, II^3^Neu5AcGg_4_Cer; (**b**) GM1-oligosaccharide chain, II^3^Neu5AcGg_4_. The GM1 sugar code is according to Varki et al. [30].

**Figure 2 ijms-21-02858-f002:**
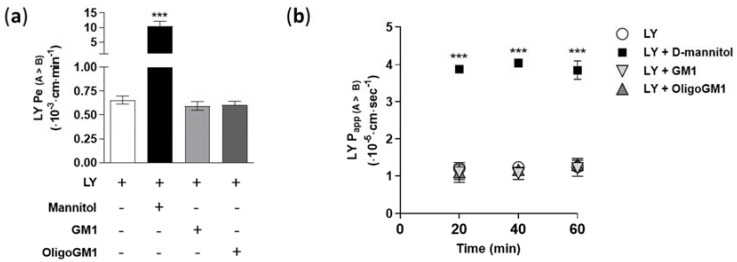
LY permeability coefficients through the hBLEC monolayer. (**a**) Direct transport-related Pe values (A → B) (×10^−3^ cm/min) of 50 μM LY alone or in combination with 50 μM GM, 50 μM OligoGM1, or 1.4 M d-mannitol after 60 min. Comparable data obtained from 10, 100, and 300 μM GM1 or OligoGM1 are shown in Appendix A. Results are mean ± SEM from 4 independent experiments, which examined a minimum of 3 wells for each condition (*** *p* < 0.01 vs. LY, LY plus GM1, and LY plus OligoGM1, two-way ANOVA, *n* = 4); (**b**) Direct transport-related Papp values (A → B) (×10^−5^ cm/s) of 50 μM LY alone or in combination with 50 μM GM1, 50 μM OligoGM1, or 1.4 M d-mannitol at different time points (20, 40, and 60 min) of hBLEC incubation. Results are mean ± SEM from 4 independent experiments (*n* = 4), which examined a minimum of 3 wells for each condition (*** *p* < 0.01 vs. LY, LY plus GM1, and LY plus OligoGM1, two-way ANOVA, *n* = 4).

**Figure 3 ijms-21-02858-f003:**
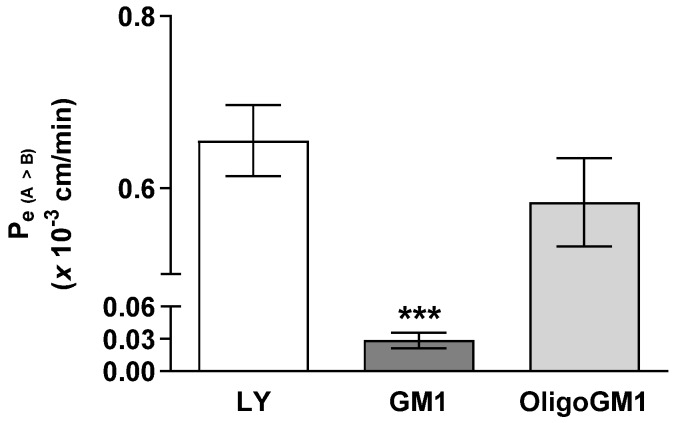
GM1 and OligoGM1 Pe coefficients through the hBLEC layer. Direct transport-related Pe (A → B) (×10^−3^ cm/min) of 50 μM OligoGM1, GM1, and LY after 60 min. Results are mean ± SEM from 4 independent experiments (*** *p* < 0.01 vs. LY and OligoGM1, two-way ANOVA, *n* = 4) performed by examining a minimum of 3 wells for each condition.

**Figure 4 ijms-21-02858-f004:**
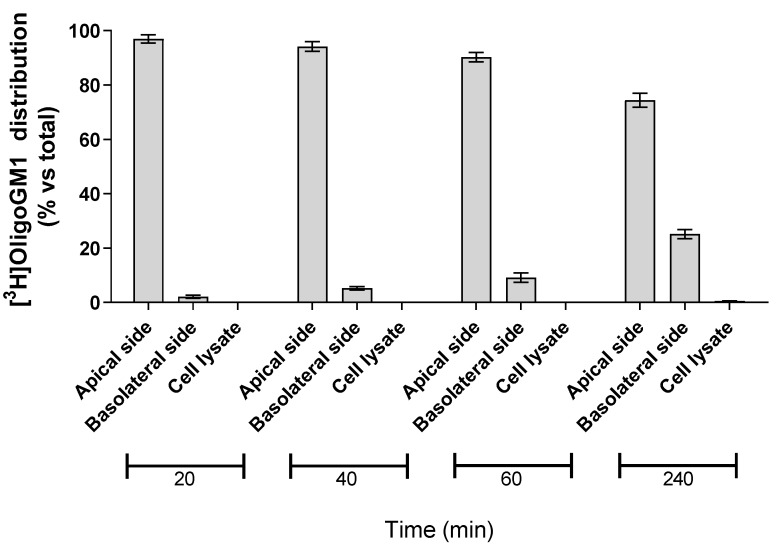
Association of [^3^H]OligoGM1 to hBLECs. Direct transport experiment (A → B) was performed as reported in the Methods section by adding 50 μM [^3^H]OligoGM1 to the apical compartment. After 20, 40, 60, and 240 min, the solutions from apical and basolateral sides were collected and hBLECs were treated to obtain the cell lysate. The radioactivity associated with each fraction was determined by liquid scintillation counting. Data are expressed as percentage mean of total radioactivity ± SEM of three different experiments (*n* = 3).

**Figure 5 ijms-21-02858-f005:**
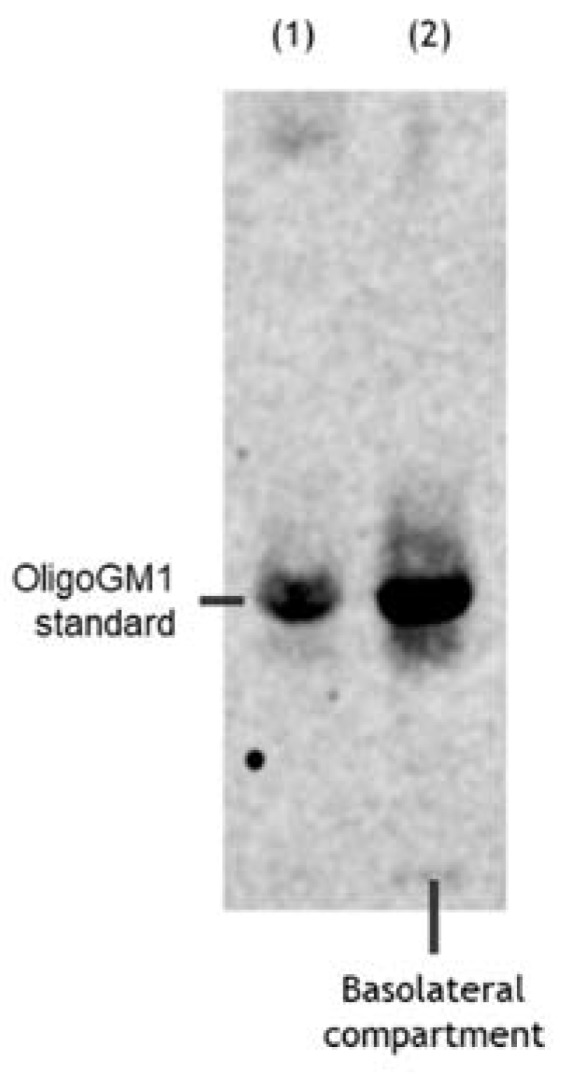
OligoGM1 stability after hBLEC transport. Representative image of the HPTLC separation of the radioactive material contained in the basolateral compartment (*n* = 4). Lane 1: standard [^3^H]OligoGM1; lane 2: concentrated solution from basolateral compartment derived from direct transport experiment with 50 μM and 1 × 10^6^ dpm [^3^H]OligoGM1 after 240 min. The result is representative of those obtained for all the experiments. HPTLC plate was developed using the chloroform/methanol/0.2% CaCl_2_ solvent system at 30:50:13 by volume. Tritium was detected with the Beta-Imager 2000 instrument (Biospace) using an acquisition time of 17 h.

**Figure 6 ijms-21-02858-f006:**
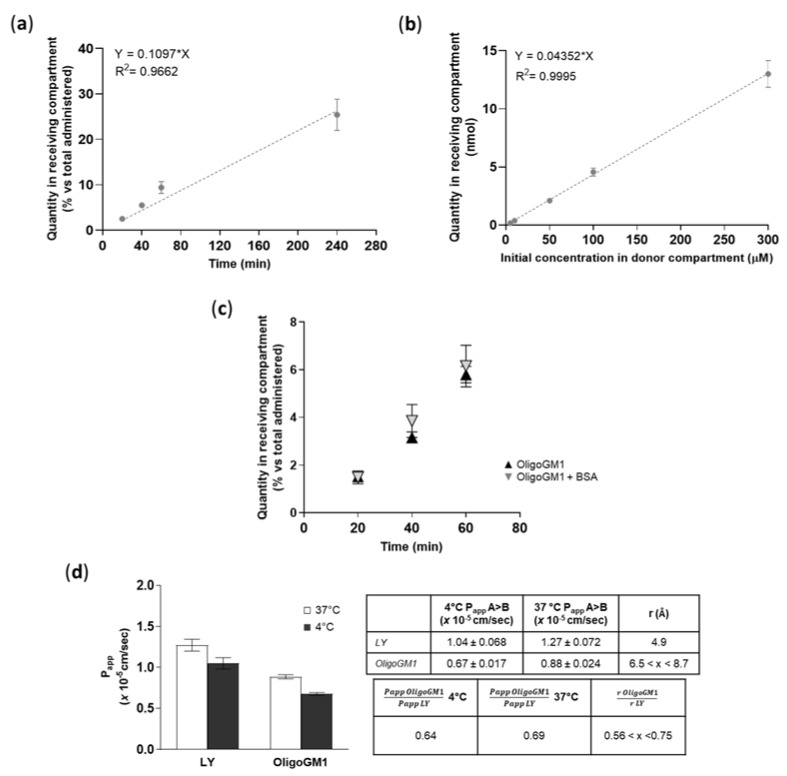
OligoGM1 transport characterization across hBLECs. (**a**) Effect of time on OligoGM1 direct transport experiment (A → B). Here, 50 μM plus 1 × 10^6^ dpm OligoGM1 was added to the apical compartment and the related quantity of OligoGM1 found in the receiving compartment after 20, 40, 60, and 240 min was measured. Data are expressed as nmol in the receiving compartment versus nmol administered (%, mean ± SEM), derived from 4 different experiment examining a minimum of 3 wells for each condition (*n* = 4). The regression model was fitted with a linear equation [y = 0.1097 *× x*], while Pearson’s correlation coefficient (r^2^) = 0.9662. (**b**) Effect of dose on OligoGM1 direct transport experiment (A → B). Here, 5, 10, 50, 100, or 300 μM plus 1 × 10^6^ dpm OligoGM1 was added to the apical compartment and the related quantity of OligoGM1 found in the receiving compartment after 60 min was measured. Data are expressed as nmol in the receiving compartment (mean ± SEM), derived from 4 different experiments examining a minimum of 3 wells for each condition (*n* = 4). The regression model was fitted with a linear equation [y = 0.04352 *× x*], while Pearson’s correlation coefficient (r^2^) = 0.9995. (**c**) Influence of BSA on OligoGM1 direct transport experiment (A → B). Here, 50 μM plus 1 × 10^6^ dpm OligoGM1 was added to the apical compartment in the absence (▲) or presence (▼) of 0.1% BSA (*w*/*v*), and the related quantity of OligoGM1 found in the receiving compartment after 20, 40 and 60 min was measured. Data are expressed as nmol in the receiving compartment versus nmol administered (%, mean ± SEM), derived from 4 different experiments examining a minimum of 3 wells for each condition (*n* = 4). (**d**) Effect of temperature on OligoGM1 direct transport experiment (A → B). Here, 50 μM LY or 50 μM plus 1 × 10^6^ dpm OligoGM1 was added to the apical compartment. After 60 min, the Papp values (×10^−5^ cm/s) were calculated. Left: Papp of LY and OligoGM1 measured after 60 min of hBLECs incubation at 37 °C (□) or 4 °C (■). Results are mean ± SEM from 4 independent experiments (*n* = 4) performed by examining a minimum of 3 wells for each condition. Right: Table indicating the ratio between Papp values at 4 °C or 37 °C and the ratio of the hydrodynamic radius (r) for LY and OligoGM1.

**Figure 7 ijms-21-02858-f007:**
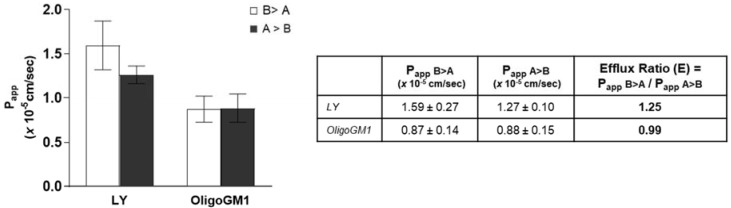
Bidirectional transport experiment across hBLECs. Left: Papp of LY and OligoGM1. For the inverse transport experiment (B > A), 50 μM LY or 50 μM plus 1 × 10^6^ dpm OligoGM1 was added to the basolateral compartment and the related quantity of OligoGM1 found in the apical compartment was measured after 60 min (□). For the direct transport experiment (A > B), 50 μM LY or 50 μM plus 1 × 10^6^ dpm OligoGM1 was added to the apical compartment and the related quantity of OligoGM1 found in the basolateral compartment was measured after 60 min (■). Right: Permeability efflux ratio (E) values for 50 μM LY and OligoGM1. Results are mean ± SEM from 4 independent experiments (*n* = 4), performed by examining a minimum of 3 wells for each condition.

**Figure 8 ijms-21-02858-f008:**
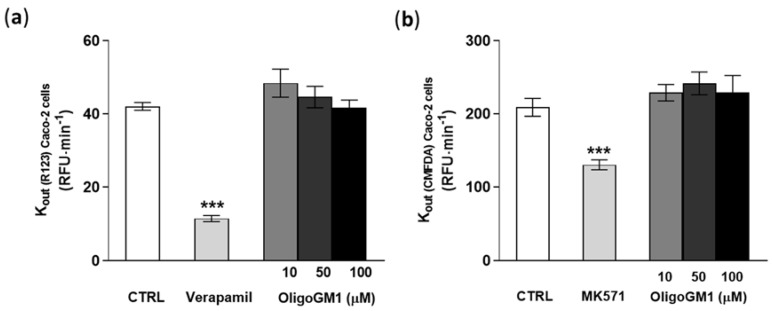
Effect of OligoGM1 on rate of excretion of rhodamine 123 (R123) (**a**) and of CMFDA (**b**). The amount of R123 or CMFDA expelled from the P-gp/BCRP or MRP transporter was calculated as the slope of the curve of the cumulative amount of dye with respect to time (Kout = ∆pmol⁄∆t) and compared with the value in the absence (CTRL) or in the presence of OligoGM1 at different concentrations (10, 50 and 100 μM). Verapamil (P-gp/BCRP competitive transport inhibitor) or MK571 (MRP competitive transport inhibitor) was used as a positive control for P-gp/BCRP or MRP efflux pumps, respectively. Data are expressed as mean ± SEM from 12 independent experiments (*** *p* < 0.01, Mann–Whitney test, *n* = 12).

**Figure 9 ijms-21-02858-f009:**
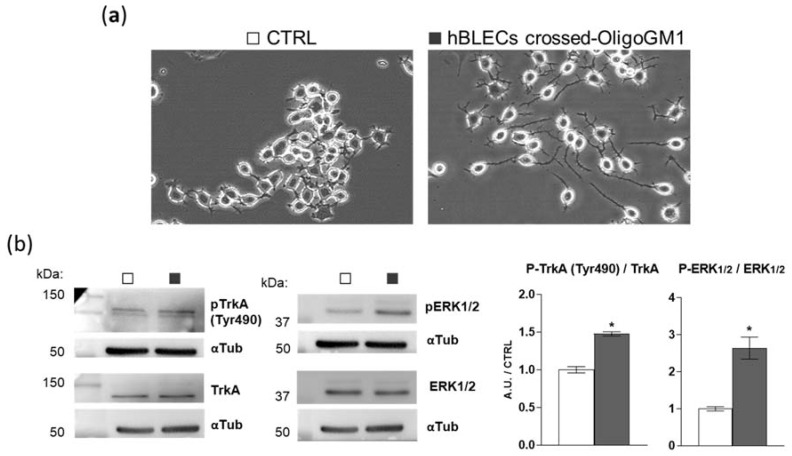
The biological activity of hBLEC-crossed OligoGM1. (**a**) Effect of hBLEC-crossed OligoGM1 on the morphology of N2a cells. After 24 h incubation, cells were analyzed with phase contrast microscopy with 200× magnification. Images are representative of 4 independent experiments (*n* = 4). N2a cells in the absence (CTRL, □) or in the presence of hBLEC-crossed OligoGM1 (■). (**b**) Effect of hBLEC-crossed OligoGM1 on the TrkA-ERK1/2 pathway. Expression of TrkA, phosphorylated TrkA (pTrkA, tyrosine 490, Tyr490), total extracellular signal-regulated protein kinase 1 and 2 (ERK1/2), phosphorylated ERK1/2 (pERK1/2, Thr202/Thr204), and α-tubulin (αTub) in cell lysate by means of specific antibodies and reveled by enhanced chemiluminescence. Left: immunoblotting images are shown. Right: semi-quantitative analysis of pTrkA, pERK1/2, TrkA, and ERK1/2 related to α-tubulin, used as internal normalizer. Data are expressed as the mean ± SEM of fold increase over control from 4 independent experiments (* *p* < 0.05, Mann–Whitney test, *n* = 4).

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
