# Peer review of "GM1 Oligosaccharide Crosses the Human Blood–Brain Barrier In Vitro by a Paracellular Route"

_ijms, 2020, doi:10.3390/ijms21082858_

Round 1

Reviewer 1 Report

In this manuscript, Di Biase et al. use a simplified BBB model to evaluate the ability of oligoGM1, an hydrophilic GM1-oligosaccharide head of the GM1 ganglioside, to enter the CNS and act as a nurotrophic, and neuroprotective agent. The authors employ an array of properly designed biological experiments to evidence that oligoGM1 is able, differently from the parent molecule GM1 ganglioside, to cross BBB via a paracellular route, without being inactivated and metabolized. The topic is of high interest due to the possible use in the design of effective AD therapies. The manuscript is well written and organized. Experimental methods are described in detail. The whole of the results fully supports the conclusions.

I recommend publication of this manuscript in the present form

minor point

Figure S1 in supplementary info appears as a table and not as a figure

Author Response

Revision according to the comments of Reviewer #1:

 1. Figure S1 in supplementary info appears as a table and not as a figure.

We modified “Figure S1” in “Table S1” along the main manuscript and in the supplementary file.

Reviewer 2 Report

The authors investigated the GM1-oligosaccharide brain transport by using a human in vitro blood brain barrier (BBB) model and assayed its metabolic integrity and functional stability. The GM1-oligosaccharide had a twenty-fold higher crossing rate than GM1 ganglioside in time and concentration dependent manners, in the passive-paracellular route and without direct interaction with the active ABC-transporters. The biological activity of GM1-oligosaccharide remained intact to induce Neuro2a cells neuritogenesis by activating TrkA pathway. The authors concluded that highly BBB-permeable GM1-oligosaccharide is more suitable for the treatment of neurodegenerative diseases.

The authors are expected to reconsider the following:

  1. Line 17: Please clarify “GM1”, “Ganglioside GM1”, “GM1 ganglioside”, “GM1-oligosaccharide”, and “GM1 oligosaccharide”.
  2. Line 79: Please define the abbreviation clearly. OligoGM1.
  3. Line 217: The equation should be shown in Methods.
  4. Line 279-290: The procedures should be included in Methods.
  5. Line 311-327: The procedures should be included in Methods.
  6. Line 347-354: This is mentioned in Introduction. Please avoid repetition.
  7. Line 347-408: Please citing more relevant literature to extend discussion, which may include arguments on historical background, drug development, advantage, disadvantage, and among others.
  8. Line 409-741: Methods should be more selective and concise. The subcategory should be arranged as presented in Results.
  9. Line 612-661: The description of calculation can be made shorter by referring corresponding literature. Please balance the proportion of Results, Discussion, and Material and Methods.
  10. Please rephrase sentence(s), choose appropriate words and correct font sizes: Lines 43-44, 46-47, 49-50, 50-52, 52-53, 54, 61-63, 64-65, 65-68, 86. 87-88, 107, 109, 157, 167-169, 170-175, 212-216, 411-412, 441, 563, 575.

The study is well-planned, well-conducted and comprehensive. Information present in this work is of great value complementing the authors’ previous research and advancing the field of GM1-oligosaccharide research. However, there are many English phrases, disproportional section length, and hesitantly developed arguments, which obscure the great value of this research article.

I strongly recommend this manuscript for publication, but after major corrections.

Author Response

Revision according to the comments of Reviewer #2:

1. Line 17: Please clarify “GM1”, “Ganglioside GM1”, “GM1 ganglioside”, “GM1-oligosaccharide”, and “GM1 oligosaccharide”.

The GM1 ganglioside (GM1) and GM1-oligosaccharide (OligoGM1) abbreviations have been corrected in the abstract section and along the manuscript.

2. Line 79: Please define the abbreviation clearly. OligoGM1.

The OligoGM1 abbreviation have been defined clearly as “GM1-oligosaccharide, II3Neu5Ac-Gg4, β-Gal-(1-3)-β-GalNAc-(1-4)-[α-Neu5Ac-(2-3)]-β-Gal-(1-4)-β-Glc, OligoGM1”.

3. Line 217: The equation should be shown in Methods. 

Initially, we reported the equation (5) of the Methods section also in the Results section to better guide the reader in understanding the discussion about the relationship between the apparent permeabilities of two molecules (LY and OligoGM1) and their hydrodynamic rays.

Following the reviewer’s suggestion, the equation has been moved to the Supplementary Information file and removed from the Results section.

4. Line 279-290: The procedures should be included in Methods. 

We modified the text reported in line 279-290 removing the procedures description which are included in the methods section.

5. Line 311-327: The procedures should be included in Methods.

We modified the text reported in line 311-327 removing the procedures description which are included in the methods section.

6. Line 347-354: This is mentioned in Introduction. Please avoid repetition.

We removed the sentences in the line 347-354.

7. Line 347-408: Please citing more relevant literature to extend discussion, which may include arguments on historical background, drug development, advantage, disadvantage, and among others.

The discussion has been carefully re-edited augmenting GM1 historical backgrounds, drug development, advantage and disadvantage.

8. Line 409-741: Methods should be more selective and concise. The subcategory should be arranged as presented in Results.

The Methods section has been shortened and subcategory has been now presented as arranged in the Results.

9. Line 612-661: The description of calculation can be made shorter by referring corresponding literature. Please balance the proportion of Results, Discussion, and Material and Methods.

The description of calculation has been edited and the original description have been moved in Supplementary section.

We agree with the reviewer in considering that the description of the formulas makes the text too long, but we prefer to present it in full as it could be useful for readers who are not expert in understanding our results.

10. Please rephrase sentence(s), choose appropriate words and correct font sizes: Lines 43-44, 46-47, 49-50, 50-52, 52-53, 54, 61-63, 64-65, 65-68, 86. 87-88, 107, 109, 157, 167-169, 170-175, 212-216, 411-412, 441, 563, 575.

The text has been edited as suggested.

Round 2

Reviewer 2 Report

The manuscript has been greatly improved by rephrasing English sentences, detailing Materials and Methods in Supplementary Files, and developing Discussion with historical backgrounds and significance in GM1-oligosaccharide research. Please reconsider the following parts:

  1. Line 19, Line 36, Lines 40-41, Line 106, Line 424, Line 492, and Line 493: Please be mindful of consistency in the nomenclature. Ganglioside GM1, GM1 ganglioside, Ganglioside GM1 (GM1, II3Neu5Ac-Gg4Cer, β-Gal-(1-3)-β-GalNAc-(1-4)-[α-Neu5Ac-(2-3)]-β-Gal-(1-4)-β-Glc-Cer), Ganglioside GM1, GM1 ganglioside, Ganglioside GM1, GM1 ganglioside.
  2. Line 22: Please place an abbreviation after “GM1-oligosaccharide” (OligoGM1) and replace the term with the abbreviation thereafter.
  3. Line 46: Please rephrase it. “…to functional recover…”. Maybe “…functional…”?
  4. Line 48: Please rephrase it. “…the possibility to … potential in clinic relies on…”.
  5. Line 50: Maybe “…GM1’s…”?
  6. Lines 61-62: Please rephrase it. “…has been observed in the rodent models has not been…”.
  7. Lines 66-69: Please rephrase it.
  8. Lines 72-74: Please rephrase it.
  9. Lines 77-81: Please rephrase it. Maybe “efforts” instead of “forces”?
  10. Lines 85-86: Would it better, “…its oligosaccharide portion, GM1-oligosaccharide, (OligoGM1) [22-25]”?
  11. Lines 92, 102, and 516: Please use the abbreviation, OligoGM1.
  12. Line 94: Please rephrase it to avoid redundancy.
  13. Lines 110 and 417: “hBLECs” is defined in Line 98.
  14. Lines 99-100: Please rephrase it “…contemplate for OligoGM1 a paracellular…”.
  15. Line 118: Maybe “Furthermore,” is better than “Following”?
  16. Lines 872-1159: Please correct reference format according to Instructions for Authors: “1. Author 1, A.B.; Author 2, C.D. Title of the article. Abbreviated Journal NameYearVolume, page range.” No comma after authors.

The study is well-planned, well-conducted and comprehensive. Information present in this work is of great value complementing the authors’ previous research and advancing the field of GM1-oligosaccharide research.

I strongly recommend this manuscript for publication, but after minor corrections.

Author Response

1. Line 19, Line 36, Lines 40-41, Line 106, Line 424, Line 492, and Line 493: Please be mindful of consistency in the nomenclature. Ganglioside GM1, GM1 ganglioside, Ganglioside GM1 (GM1, II3Neu5Ac-Gg4Cer, β-Gal-(1-3)-β-GalNAc-(1-4)-[α-Neu5Ac-(2-3)]-β-Gal-(1-4)-β-Glc-Cer), Ganglioside GM1, GM1 ganglioside, Ganglioside GM1, GM1 ganglioside.

The term Ganglioside has been repositioned to always be before the term GM1.

 2. Line 22: Please place an abbreviation after “GM1-oligosaccharide” (OligoGM1) and replace the term with the abbreviation thereafter.

The GM1-oligosaccharide abbreviation has been corrected has suggested.

3. Line 46: Please rephrase it. “…to functional recover…”. Maybe “…functional…”?

The text has been rephrased as suggested.

4. Line 48: Please rephrase it. “…the possibility to … potential in clinic relies on…”.

The text has been rephrased as suggested.

 5. Line 50: Maybe “…GM1’s…”?

The text has been edited as suggested.

 6. Lines 61-62: Please rephrase it. “…has been observed in the rodent models has not been…”.

The text has been rephrased as suggested.

7. Lines 66-69: Please rephrase it.

The text has been rephrased as suggested.

 8. Lines 72-74: Please rephrase it.

The text has been rephrased as suggested.

9. Lines 77-81: Please rephrase it. Maybe “efforts” instead of “forces”?

The text has been rephrased as suggested.

 10. Lines 85-86: Would it better, “…its oligosaccharide portion, GM1-oligosaccharide, (OligoGM1) [22-25]”?

The text has been edited as suggested.

11. Lines 92, 102, and 516: Please use the abbreviation, OligoGM1. 

The abbreviation has been used correctly now.

12. Line 94: Please rephrase it to avoid redundancy.

The text has been rephrased as suggested.

13. Lines 110 and 417: “hBLECs” is defined in Line 98. 

The text has been edited as suggested.

14. Lines 99-100: Please rephrase it “…contemplate for OligoGM1 a paracellular…”.

The text has been rephrased as suggested.

15. Line 118: Maybe “Furthermore,” is better than “Following”? 

“Following” has been substituted with “Furthermore” as suggested.

16. Lines 872-1159: Please correct reference format according to Instructions for Authors: “1. Author 1, A.B.; Author 2, C.D. Title of the article. Abbreviated Journal NameYear, Volume, page range.” No comma after authors.

The references have been formatted according to Authors’ instruction.